# Lyophilization Serves as an Effective Strategy for Drug Development of the α9α10 Nicotinic Acetylcholine Receptor Antagonist α-Conotoxin GeXIVA[1,2]

**DOI:** 10.3390/md19030121

**Published:** 2021-02-25

**Authors:** Zhiguo Li, Xiaolu Han, Xiaoxuan Hong, Xianfu Li, Jing Gao, Hui Zhang, Aiping Zheng

**Affiliations:** Department of Pharmaceutics, Institute of Pharmacology and Toxicology, Academy of Military Medical Sciences, 27th Taiping Road, Haidian District, Beijing 100850, China; ziguolee@163.com (Z.L.); hanxiaolu921007@163.com (X.H.); hongxiaoxuan1216@163.com (X.H.); xiaofu0924@163.com (X.L.); gjsmmu@126.com (J.G.); zhhui58@126.com (H.Z.)

**Keywords:** α-Conotoxin GeXIVA[1,2], nicotinic acetylcholine receptors, lyophilization, molecular simulation, paclitaxel-induced neuropathic pain, analgesic effect

## Abstract

α-Conotoxin GeXIVA[1,2] is a highly potent and selective antagonist of the α9α10 nicotinic acetylcholine receptor (nAChR) subtype. It has the advantages of strong efficacy, no tolerance, and no effect on motor function, which has been expected help patients with neuropathic pain. However, drug development for clinical use is severely limited owing to its instability. Lyophilization is applied as the most preferred method to solve this problem. The prepared lyophilized powder is characterized by differential scanning calorimetry (DSC), powder X-ray diffractometry (PXRD), and Fourier transform infrared spectroscopy (FTIR). Molecular simulation is also used to explore the internal distribution and forces formed in the system. The analgesic effect on paclitaxel-induced neuropathic pain following single and 14-day repeated administrations are evaluated by the von Frey test and the tail-flick test. Trehalose combined with mannitol in a ratio of 1:1 is employed as the excipients in the determined formulation, where trehalose acts as the stabilizer and mannitol acts as the bulking agent, according to the results of DSC, PXRD, and FTIR. Both GeXIVA[1,2] (API) and GeXIVA[1,2] lyophilized powder (formulation) could produce stable analgesic effect. These results indicated that GeXIVA[1,2] lyophilized powder could improve the stability and provide an effective strategy to push it into clinical use as a new analgesic drug.

## 1. Introduction

Neuropathic pain caused by injury or dysfunction in the peripheral and central nervous system remains to be a global problem, which leads to decreases in the physical and mental quality of people’s life [1]. There is still a limitation of effective clinical pain management, due to the lack of highly specific analgesics. Currently, the most effective analgesics are opioids which exploit an endogenous pain control pathway within the central nervous system in the imperfect available analgesics [2,3]. Whereas, opioids have significant issues with tolerance, dependence, respiratory depression, and opioid-induced hyperalgesia [4,5,6]. To find new treatments of neuropathic pain, some new targets have been searched, including ion channels [7], transduction molecules [8], and nicotinic acetylcholine receptors (nAChRs) [9,10]. The nAChRs are related to a number of diseases, including pain, Alzheimer’s and Parkinson’s disease, small cell lung carcinoma, nicotinic addiction, schizophrenia, and attention-deficit hyperactivity disorder, etc. [11,12,13,14,15,16]. Targeting specific relevant nAChR subtypes may be an attractive pharmaceutical strategy to develop better analgesic drugs. The α9α10 nAChR subtype is a recently identified target for the development of breast cancer chemotherapeutics and analgesics, particularly to treat neuropathic pain [17,18,19,20].

Conotoxins could selectively combine with a large variety of nAChR subunit assemblies, including α9α10 nAChR. So far, several potential novel analgesics have been identified [21,22,23]. αO-conotoxin GeXIVA was recently identified and found to be the antagonist of α9α10 nAChR [24]. This peptide is composed of 28 amino acids with four Cys residues that can form three different disulfide bond connection isomers, i.e., GeXIVA[1,2] (bead), GeXIVA[1,3] (globular), and GeXIVA[1,4] (ribbon). Among them, GeXIVA[1,2] was the most potent antagonist of α9α10 nAChR with the IC50 of 4.6 nM [24]. GeXIVA[1,2] exhibits higher or similar potency at rat α9α10 nAChR comparing with Vc1.1 [25] or RgIA [26]. As a new candidate analgesic drug, GeXIVA[1,2] has the advantages of strong efficacy, no tolerance, and no effect on motor function [27]. It is expected to bring a new strategy for patients suffering from neuropathic pain. However, peptides, including conotoxins, typically have low oral bioavailability and are easily degraded by proteases in vitro and in vivo, thus decreasing their therapeutic potential. Unfortunately, the instability of GeXIVA[1,2] would severely limit its drug development for clinical application [28].

Lyophilization or freeze-drying is the most preferred method to stabilize biopharmaceuticals like proteins and peptides [29,30,31]. The expansion of freeze-drying application into the pharmaceutical field is largely based on the understanding that stability and viable shelf-life may be achieved with a significant reduction of moisture content [32]. Besides, the whole process is performed at a lower temperature which would significantly improve the stability of peptides which are marginally stable under thermal conditions. However, some peptides would be inactivated during the lyophilization process, due to various stresses or degrade during the storage stage [33]. Some degraded peptides would not only lose their therapeutic effects, but also cause side effects like local inflammation, mechanical disruption to tissues, or immune responses to the formulation [34,35]. Therefore, certain excipients, also called cryoprotectants, would be added to protect the peptides against physical stress associated with the free-drying process and storage condition. Some excipients embed the active ingredients in glass-state solids with low molecular mobility, thereby reducing chemical reactivity [36,37]. Some excipients provide protective effects by substituting the molecular interactions provided by water molecules. Thus, the process control and usage of appropriate excipients are quite important to protect peptides during lyophilization [36].

In our present study, the stability of GeXIVA[1,2] was investigated, and the formulation of GeXIVA[1,2] lyophilized powder was developed with a suitable process and optimized prescription. The freeze-dried samples were characterized by differential scanning calorimetry (DSC), powder X-ray diffractometry (PXRD), and Fourier transform infrared spectroscopy (FTIR). Molecular simulation was also developed to understand the internal structure and interactions between GeXIVA[1,2] and excipients of the formulation. Two common pain measurement methods, the von Frey test and the tail-flick test, were utilized to test the pain reliever effect of the lyophilized formulation by single administration and repeated treatments.

## 2. Results

### 2.1. Stability Study of GeXIVA[1,2] under Different Conditions

As shown in Table 1, in a neutral environment and an acidic medium of 0.1 M HCl, the remaining amount of GeXIVA[1,2] were 98.74 ± 0.12% and 97.59 ± 0.23%, respectively. However, GeXIVA[1,2] was extremely unstable in the alkali medium, with the assay decreased to 88.22 ± 0.95% during only 2 min incubation in 0.1 M NaOH. In an oxidative condition of 3% H_2_O_2_ (to shorten the destruction time), the remaining concentration of GeXIVA[1,2] was 89.35 ± 0.48%, which also demonstrated that oxidation could affect the stability of GeXIVA[1,2]. Exposing GeXIVA[1,2] at 4500 Lx (fluorescent) for 5 days and 10 days, the remaining amount were 97.46 ± 0.21% and 95.37 ± 0.32%, respectively. The remaining amounts of GeXIVA[1,2] at 40 °C and 60 °C for five days were 80.53 ± 0.98% and 23.79 ± 1.42%, respectively, indicating that temperature significantly affected the stability of GeXIVA[1,2], especially in a high-temperature environment. More than 95% of the original amount remained in buffers with the pH of 4.0, 5.0, 6.8, and 7.4 during the 24 h incubation. It demonstrated that GeXIVA[1,2] was comparatively stable within a pH range of 4.0–7.4. The percentages of the remaining amount of GeXIVA[1,2] in alkaline solution with pH of 8.0 and 9.0 were much lower than that in the acid buffer, which meant that GeXIVA[1,2] might be comparatively susceptive to alkaline environments.

### 2.2. Preparation of GeXIVA[1,2] Lyophilized Powder for Injection

The appearances and assay of GeXIVA[1,2] lyophilized products are the two main ingredients to evaluate the effect of cryoprotectant employed in the final formulation. Good appearances can ensure the acceptability of patients in clinical use, and the assay is the critical quality to ensure the efficacy of drug products. Except for the presence of incompatible excipients, there are another two factors that may cause the change of GeXIVA[1,2] assay, one is the freeze-drying process, due to the freeze stress and drying stress [37], the other is the storage process with the change of temperature, light, and humidity [38]. Therefore, both the two processes are also concerned.

As demonstrated in Figure 1 (left), when a single cryoprotectant was applied in the formulation, trehalose, glucose, and sorbitol produced poor appearances, while mannitol, glycine, and polyethylene glycol 2000 (PEG 2000) showed relatively perfect appearances. The protective effects of using different single cryoprotectants on assay were shown in Table 2. In detail, trehalose, mannitol, sorbitol, and glycine had good protective effects on the molecule structure of GeXIVA[1,2] during the freeze-drying process. However, for the storage stability, only trehalose had good maintenance of the assay of GeXIVA[1,2] after storage for 30 days under accelerating conditions (40 °C, 75% RH). In conclusion, trehalose might not be a good bulking agent, but acted as a good stabilizer.

The appearances of different formulations obtained by using combinations of two cryoprotectants in the 1:1 ratio were displayed in Figure 1 (right). The appearances of the combinations of trehalose and glucose/sorbitol are not qualified. However, when trehalose was combined with mannitol, glycine, and PEG 2000, all formulations produced excellent appearances. According to the assay results, shown in Table 2, the combination of trehalose with another cryoprotectant could effectively maintain the assay of GeXIVA[1,2] during the freeze-drying process, except for the lactose. In terms of storage stability, only when trehalose was combined with mannitol, it could maintain the assay of GeXIVA[1,2] during 30 days of storage under accelerating conditions.

In summary, the combination of trehalose and mannitol applied in the formulation could produce a perfect appearance for the product, and have good maintenance of the assay. Therefore, trehalose and mannitol were selected as the best combination in the determined formulation, where trehalose acted as the stabilizer and mannitol as the bulking agent.

### 2.3. Characterization of GeXIVA[1,2] Lyophilized Powder for Injection

#### 2.3.1. Proprieties of the Lyophilized Products

The lyophilized products exhibited a pharmaceutically elegant cake structure with uniform size and shape, and no obvious difference between and within batches was found. Low moisture content is an essential criterion for lyophilized products. According to previous studies, the moisture content in lyophilized samples is commonly less than 3% (*w*/*w*) [39,40]. The lyophilized products were found to have a low moisture content of 1.89 ± 0.08%, which was within the limit. When reconstituting the GeXIVA[1,2] lyophilized powder, the samples were completely reconstituted within 10 s at room temperature without shaking. The solution was clear and colorless after reconstitution, and no insoluble particles were observed under the light. The osmotic pressure of the solution reconstituted by water was 130 mOsm/kg, which was hypotonic, while reconstituted by normal saline was 418 mOsm/kg, which was almost isotonic. Based on the guideline of clinical use of lyophilized powder for injection, the range of 0.5–2 times of normal saline osmotic pressure for intramuscular (IM) injection is allowed [41]. Therefore, normal saline is recommended for the reconstitution of GeXIVA[1,2] lyophilized powder in clinical use. The acidity of the solution after reconstitution was around pH 6.0, which was helpful to reduce the irritation of intramuscular injection.

#### 2.3.2. Differential Scanning Calorimetry (DSC)

Differential scanning calorimetry (DSC) is widely used to observe the thermal behavior of different samples. The heat flow can reflect the enthalpy change of the sample. In detail, when the sample absorbs energy, the enthalpy becomes endothermic; when the sample releases energy, the enthalpy becomes exothermic. In the DSC curve, the thermal effects, such as melting, crystallization, solid-solid phase transition, and chemical reaction, are peak-shaped; for the specific heat capacity changes, such as glass transition, they are step-shaped [42,43]. DSC thermograms of the samples were displayed in Figure 2. The pure GeXIVA[1,2] exhibited an amorphous state (Figure 2a). The pure trehalose and mannitol showed a crystalline state with sharp melting endothermic peaks at 101.24 °C and 168.06 °C, respectively (Figure 2b,c). When trehalose and mannitol were lyophilized together, the melting endothermic peaks of the two substances both decreased, indicating some interaction happened (Figure 2d). When trehalose was applied singly in the formulation, the freeze-dried sample also showed an amorphous state (Figure 2e). While mannitol was used singly (Figure 2f), mannitol was in the crystalline state in the lyophilized samples. But an additional small peak appeared, which may be caused by the interaction between mannitol and GeXIVA[1,2]. In the determined formulation (Figure 2g), the thermogram was almost the same with trehalose and mannitol lyophilized together (without GeXIVA[1,2]), which meant the interaction between trehalose and mannitol still existed, but that between GeXIVA[1,2] and mannitol disappeared.

#### 2.3.3. Powder X-ray Diffractometry (PXRD)

Powder X-ray diffractometry (PXRD) analysis is an effective method for detecting formulations in crystalline and amorphous states. The diffractogram of pure GeXIVA[1,2] had no high-intensity sharp peaks, which meant it was in the amorphous state (Figure 3a). In contrast, pure trehalose and pure mannitol both displayed numerous high-intensity sharp peaks, indicating the presence of crystalline (Figure 3b,c). When trehalose and mannitol were lyophilized together, the height of peaks changed, and some peaks disappeared, meaning the crystalline state of trehalose and mannitol had been changed after freeze-drying (Figure 3d). When GeXIVA[1,2] lyophilized with different excipients, different states were observed. Specifically, when trehalose was applied singly, the sample displayed an amorphous state similar to pure GeXIVA[1,2] (Figure 3e). However, it showed a crystalline state when mannitol was singly used, even some peaks decreased comparing to pure mannitol (Figure 3f). For the determined formulation (Figure 3g), trehalose combined with mannitol was used as the cryoprotectants to lyophilize with GeXIVA[1,2]; the peaks were similar to that of mannitol combining with trehalose (without GeXIVA[1,2]). The results of Figure 3d,f,g meant that the mannitol showed the dominant crystalline state in the lyophilized system.

#### 2.3.4. Fourier Transform Infrared Spectroscopy (FTIR)

Fourier transform infrared (FTIR) spectroscopy is one of the major techniques of structural characterization of peptides and peptide-excipients interactions. Changes in amide bonds may have significant impacts on the activity and content of peptides. Thus, the characteristic peaks of amide bonds of GeXIVA[1,2] in different formulations were paid more attention. The pure GeXIVA[1,2] showed two characteristic peaks at 1656 cm^−1^ and 1542 cm^−1^, which belong to the amide band I and amide band II, respectively (Figure 4a). In the formulation where trehalose singly applied as the cryoprotectant, the two peaks had a little shift to 1660 cm^−1^ and 1552 cm^−1^, respectively. The amide bonds were still remained, indicating that trehalose had no obvious effect on the structure of GeXIVA[1,2] and could retain its activity (Figure 4d). The peak of amide band I did not shift and maintained at 1656 cm^−1^ when mannitol was singly used as an excipient in the formulation, but the intensity was somewhat weakened (Figure 4e). In addition, the peak of amide band II disappeared, indicating that mannitol had a great impact on the amide II band of GeXIVA[1,2], which might be the reason why the assay decreased. In the determined formulation (Figure 4g), trehalose combined with mannitol as lyophilized excipients, the attribution of two amide bonds were retained and had a little shift to 1669 cm^−1^ and 1553 cm^−1^, respectively. This result suggested that the determined formulation had no obvious effect on GeXIVA[1,2] and could maintain the structural integrity, which was consistent with the results of formulation development.

### 2.4. Molecular Simulation

The molecular simulation aims to build a set of models and algorithms based on experiments or the basic principles, so as to calculate the reasonable molecular structure and molecular behavior. It can simulate not only the static structure of molecules, but also the dynamic behavior of molecular systems. In this simulation, the molecular distributions of GeXIVA[1,2], trehalose, and mannitol were characterized. The hydrogen bonds among GeXIVA[1,2] and excipients were predicted. The other interactions, including van der Waals force and electrostatic interaction, were calculated, too. Moreover, the secondary structure of GeXIVA[1,2] was concerned.

Trehalose and mannitol were evenly distributed around GeXIVA[1,2], and hydrogen bonds were formed among them according to Figure 5a. The distribution density of trehalose and mannitol arrived at a highest at the distance of 2.0 nm. The distribution density of trehalose was slightly higher than that of mannitol between 0.5 and 1.8 nm, which means that trehalose was closer around the GeXIVA[1,2] in the formulation structure (Figure 5b). Hydrogen bonds between GeXIVA[1,2] and the excipient molecules were frequently observed. In the formulation, the number of hydrogen bonds between GeXIVA[1,2] and trehalose was similar to that formed between GeXIVA[1,2] and mannitol, all fluctuating from 10 to 25. More hydrogen bonds were formed between trehalose and mannitol, ranging from 175 to 225 (Figure 5c).

According to Figure 6, the electrostatic force and van der Waals force between GeXIVA[1,2] and trehalose in the system were about −1000 kJ/mol, which was similar to that between GeXIVA[1,2] and mannitol. However, the electrostatic force and van der Waals force between trehalose and mannitol in the system was higher, fluctuating in the range of −7000~−10,000 kJ/mol and −8000~−10,000 kJ/mol, respectively. The secondary structure of GeXIVA[1,2] in the system was relatively stable and dominated mainly by α-helix, turn, and coil. Moreover, amino acid fragment 20–24 could form a 5-turn helix, that is, π helix (Figure 7). The predicting result was consistent with the physical characterization.

### 2.5. Evaluation of the Analgesic Effect of GeXIVA[1,2] Lyophilized Powder for Injection

#### 2.5.1. Development of Neuropathic Pain in Rats

Rats receiving paclitaxel developed neuropathic pain between 5–7 days post-injection, while the pain threshold of the normal saline (NS) group did not change. The mechanical paw withdrawal threshold (PWT) decreased from 15.70 ± 3.29 g (before) to 1.92 ± 0.74 g (after). For heat sensitivity, the tail-flick latency (TFL) in response to the heat stimulation decreased from 11.10 ± 1.28 s (before) to 5.79 ± 1.22 s (after). These results indicated the successful development of neuropathic pain in rats, and were similar to the results obtained in previous research [27].

#### 2.5.2. Pain Reliever Effect of GeXIVA[1,2] Lyophilized Powder by Single Administration

As shown in Figure 8, both GeXIVA[1,2] (API) group and GeXIVA[1,2] lyophilized powder (formulation) group increased mechanical PWT and TFL after a single injection, while the NS group did not change during this whole period. A single intramuscular (IM) injection of both API and formulation produced analgesia effect on paclitaxel-induced mechanical allodynia starting at 1 h and persisting to 4 h and 6 h post-injection, respectively. Heat hyperalgesia was also reversed after the first injection, and this effect was maintained from 1 h to 6 h.

#### 2.5.3. Acute Analgesic Effect of GeXIVA[1,2] Lyophilized Powder after Repeated Injections

Aiming to study the acute analgesic effect by repeated treatments, the mechanical PWT and TFL of rats in different groups were detected at 4 h post-injection during a 14-day repeated treatment procedure (Figure 9). Both API and formulation reversed the mechanical allodynia and heat hyperalgesia during the 14-day repeated treatment and displayed a significant difference from the NS group. The NS treatment did not affect the PWT and TFL at any time point during the whole experiment.

## 3. Discussion

Marine drugs are continually being paid more attention in pharmaceutical research during the 21st century. Ziconotide (ω-conotoxin MVIIA, Prialt^®^), a conotoxin drug identified from the venom of *Conus magus*, has been approved by the FDA, and is mainly used in the treatment of intractable pain in the late stage of cancer and AIDS. It has a definite curative effect and produces fewer side effects compared with opioids [44,45]. However, ziconotide exerts its analgesic effect by blocking the N-type calcium channel in the central nervous system. In this condition, intrathecal central administration is required, but the administration route is very troublesome in clinical use. Therefore, the development of new non-addictive analgesics with a more convenient administration route (such as IM injection) will have a broader market prospect.

It has been proved that α9α10 nAChR is mainly distributed in the peripheral nervous system, and is related to the transmission of pain signal [23]. Previous studies have shown that GeXIVA[1,2] could selectively acts on α9α10 nAChR and has good analgesic activity in rats model of chronic constriction injury (CCI) [18] and chemotherapy-induced neuropathic pain (CINP) [27] administrated by IM injection. Therefore, GeXIVA[1,2] can be used as a promising analgesic drug that has broad prospects in the treatment of neuropathic pain.

Like most other peptides, GeXIVA[1,2] is unstable in solution, due to the characteristics of large molecular weight, complex structure, disulfide bond, and temperature sensitivity. Therefore, there will be great risks for GeXIVA[1,2] in drug development, storage, and management.

Lyophilization is widely applied in the development of peptide drugs, due to the special advantages. However, freeze-drying is a very complex physical process containing the changes of electrolyte concentration, pH value, and temperature of storage that will affect the structure and function of peptide drugs [46,47]. Therefore, cryoprotectants play an important role in the stability of peptide drugs during the whole life cycle. They usually have the characteristics of poor water absorption, high glass transition temperature (Tg), low crystallization rate, and no reducing group [29,48]. Carbohydrate and polyhydric alcohol are the most popular cryoprotectants. Sometimes, if the single cryoprotectant could not achieve the desired effect, a combination of multiple cryoprotectants would be tried. In the present study, the combination of trehalose and mannitol was employed as the freeze-drying protectants and finally achieved quite good results.

There are many protective mechanisms of lyophilization in which glass embedding and water substitution hypotheses are widely recognized [49], and were also proved in this study. In detail, when trehalose is employed as the cryoprotectant, the freeze-drying system is amorphous, and no obvious interaction is observed between them from the results of DSC, XRD, and FTIR. As a result, even the formulation shows poor appearance, the stability of GeXIVA[1,2] in the system is good enough. The reason may be that trehalose can form a glassy state which has the behavior of both solid and fluid with high viscosity and low diffusion coefficient. Therefore, when lyophilized with GeXIVA[1,2], trehalose surrounds the GeXIVA[1,2] to hinder its movement and stabilize the structure, thus plays as a protective agent. While mannitol is singly used as the cryoprotectant, the freeze-drying system shows a crystalline state and displays a good appearance, but the stability of GeXIVA[1,2] is not qualified, due to the damage interaction. To the determined formulation, trehalose combining with mannitol as cryoprotectants, the interaction between mannitol and trehalose is obvious, but it does not affect the structure of GeXIVA[1,2], which means that trehalose preferentially distributes around GeXIVA[1,2] to avoid the damage interaction from mannitol. Finally, mannitol acts as a bulking agent to support the skeleton structure of the powder cake showing the full appearance, while trehalose provides a glassy environment for GeXIVA[1,2] to maintain the structure and stability, improving the stability of GeXIVA[1,2] during the freeze-drying and storage processes. Besides, hydrogen bonding and other weak binding forces, which are also involved in maintaining the structure and activity of GeXIVA[1,2], are observed among GeXIVA[1,2], trehalose and mannitol from the molecular simulation results. That may because when the peptide loses water in the drying process, the hydroxyl group of the trehalose and mannitol can replace the hydroxyl group of the water and form a hypothetical hydration film on the surface of GeXIVA[1,2], which can protect the bonding position of the hydrogen bonds from being directly exposed to the environment.

Alkaloids, paclitaxel, platinum, and some other anti-tumor drugs can produce neurotoxicity and induce neuropathic pain, which is characterized by hyperalgesia. This side effect caused by anti-tumor drugs is called chemotherapy-induced neuropathic pain (CINP) [50]. The three-step analgesic therapy suggested by WHO shows limited effects on CINP unless reducing the dosage of chemotherapy drugs or stopping their use [51,52]. The therapeutic analgesic effect of GeXIVA[1,2] in the established oxaliplatin-induced acute neuropathic pain model was confirmed in the previous study [27]. In this paper, we selected paclitaxel as the inducer of CINP in rats and investigated the pain reliever effect of GeXIVA[1,2] (API) and GeXIVA[1,2] lyophilized powder (formulation). On the one hand, it was evaluated whether GeXIVA[1,2] could play the analgesic effect in the CINP model induced by paclitaxel compared to the oxaliplatin-induced model. On the other hand, we explored whether the analgesic effect or biological activity of GeXIVA[1,2] was decreased when it was prepared as a lyophilized powder. Our research has shown that continuous use of paclitaxel can cause hyperalgesia in rats. The von Frey test and the tail-flick test are common pain measurement methods utilized to test the pain reliever effect, and were used in the present study. The results show that both API and formulation could produce a stable analgesic effect in the CINP model induced by paclitaxel. The formulation presents little better-analgesic effects than API, indicating the success of the formulation development. These results suggest that GeXIVA[1,2] lyophilized powder could be a promising drug for chemotherapy-induced neuropathic pain management.

## 4. Materials and Methods

### 4.1. Animals

Animals (male Sprague-Dawley rats weighing 180–200 g; Beijing animal center, Beijing, China) were housed in plastic cages and maintained under controlled conditions (a fixed 12 h light/dark cycle, temperature and humidity-controlled room, food and water available). The animals were given 3–4 days to adapt to the housing facilities and the testing procedures prior to the experiments. All animal procedures were carried out between 9:00 and 15:00 All the experiments complied with the ARRIVE guidelines [53], and were approved by the Animal Ethics Committee and the Institutional Animal Care and Use Committee of Beijing Institute of Pharmacology and Toxicology, Beijing, China (IACUC of AMMS-06-2017-001). Furthermore, all experiments were carried out following the Guidelines for the Care and Use of Laboratory Animals [54]. Rats were randomly allocated to each experiment. We made all efforts to minimize animal suffering and the number of animals used.

### 4.2. Compounds

Synthesis of GeXIVA[1,2] performed as previously described [24]. Trehalose was in injection-grade and purchased from Hayashibara Co., Ltd. (Okayama, Japan). Injection-grade glucose, mannitol, sorbitol, and glycine were purchased from Beijing Fengli Jingqiu Pharmaceutical Co., Ltd. (Beijing, China). Polyethylene glycol 2000 (PEG 2000) was purchased from BASF (Ludwigshafen, Germany). Paclitaxel was provided by Zhejiang Hisun Pharmaceutical Co., Ltd. (Taizhou, China). HPLC-grade triethylamine and acetonitrile were obtained from Thermo Fisher Scientific (Waltham, MA, USA). The purified water used in this study was prepared using a Mille-Q system (EMD Millipore, Billerica, MA, USA).

### 4.3. High-Performance Liquid Chromatography (HPLC) Analysis

The concentration of GeXIVA[1,2] was determined using an HPLC system. A reversed-phase HPLC column (Kromasil-100, 3.5 μm × 4.6 mm × 250 mm) was used for the chromatographic separations. The mobile phase consisted of acetonitrile and 0.05% triethylamine in water (14:86, *v*/*v*). Samples were eluted in isocratic mode with a flow rate of 1.0 mL/min at 40 °C. The detection wavelength was set at 215 nm, and the injection volume was 20 μL.

### 4.4. Stability Study of GeXIVA[1,2] under Forced Degradation

Forced degradation studies play an important role in the drug development process [55]. Stress conditions commonly used include acid, alkali, oxidative stress, temperature changes, light exposure, and pH levels, all of which can potentially affect peptide stability [56]. Therefore, a series of stress tests for GeXIVA[1,2] was performed based on the above conditions that may be exposed during the production, storage, and transport (Table 1).

### 4.5. Preparation of Lyophilized Formulations

Sugars (trehalose, sucrose), polyols (mannitol and sorbitol), amino acid (glycine), and polymer (PEG2000) were employed as excipients for the formulation development. In details, GeXIVA[1,2] solution (0.4%, *w*/*v*) and solutions (5%, *w*/*v*) of each excipient were prepared by dissolving in purified water. Then solutions of each single cryoprotectant or combinations of two cryoprotectants in the 1:1 ratio were mixed with GeXIVA[1,2] to obtain different desired formulations. In each formulation, the content of GeXIVA[1,2] is 0.2%, and the content of excipients is 2.5%. After filtering, 1 mL solution of different formulations was aliquoted into 3 mL medium borosilicate glass vials (Gerresheimer Shuangfeng Pharmaceutical Glass (Danyang) Co., Ltd., Zhenjiang, China) then capped with 2-legged high purity stoppers in brominated butyl rubber (Hualan Co., Ltd., Jiangyin, China), and finally lyophilized according to the freeze-drying process. The samples were frozen at an initial shelf temperature of −45 °C for 6 h. Primary drying was conducted at a chamber pressure of 0.2 millibars with shelf temperature adjusted to −10 °C and held for 10 h. Following primary drying, shelf temperature was increased to 20 °C at a ramp rate of 10 °C/h, and secondary drying was performed for about 6 h. At the end of the cycle, vials were automatically stoppered and sealed with aluminum caps. Then the different formulation products were stored under an environment of 40 °C and 75% RH as the accelerating condition. The appearances of the lyophilized products and assay changes of GeXIVA[1,2] during the freeze-drying process and storage process were recorded and compared.

### 4.6. Characterization of GeXIVA[1,2] Lyophilized Powder for Injection

#### 4.6.1. Moisture Content Determination

The moisture content of GeXIVA[1,2] lyophilized powder was measured via a Karl Fischer titration instrument (870 Titrando, Metrohm, Herisau, Switzerland). After the instrument reached equilibrium, the lyophilized powder was accurately weighed and placed into the titration chamber dissolved at 100 rpm. The moisture content was recorded, and the test was repeated three times.

#### 4.6.2. Reconstitution of GeXIVA[1,2] Lyophilized Powder

Each sample was reconstituted with 1.0 mL distilled water or normal saline, while putting the solution flow onto the inside of the vial. The time to ensure proper wetting of the lyophilized cake was recorded. The color and clarity of the solution, visible particles, osmotic pressure, and acidity were detected.

#### 4.6.3. Differential Scanning Calorimetry (DSC)

Differential scanning calorimetry (DSC) was performed in a DSC Q2000 (TA Instruments, New Castle, DE, USA) equipped with a TA instrument Universal Analysis software, an autosampler, and a cooling system. Nitrogen gas was purged at a pressure of 1.38 bar to provide an inert atmosphere and prevent oxidation during measurement. About 5 mg of lyophilized powder sample was placed in a crimped aluminum hermetic pan, and DSC was performed at a ramp rate of 5 °C/min to 200 °C. Individual components of the formulation in its pure non-lyophilized form were also scanned to obtain the thermograms for comparison.

#### 4.6.4. Powder X-ray Diffractometry (PXRD)

Powder X-ray diffractometry (PXRD) was performed using a Bruker D8 Advance with DaVinci design (Bruker AXS, Madison, WI, USA) for evaluating the lyophilized products. The X-ray source was Kα radiation from a copper target with a graphite monochromator at a wavelength of 1.54 Å. The range (2θ) of scans was 5° to 50°, with the rate of 10°/min.

#### 4.6.5. Fourier Transform Infrared Spectroscopy (FTIR)

Fourier transform infrared spectra (FTIR) were collected for lyophilized samples using a Bruker Tensor 27 (Bruker Optics, Billerica, MA, USA) equipped with an attenuated total reflection (ATR) accessory. All the samples were measured in the frequency range of 500–4000 cm^−1^ with a data density of 4 cm^−1^ and a total of 50 scans using OPUS software (Bruker Optics, version 7.0).

### 4.7. Molecular Computations

The generation amber force field (GAFF) was used for trehalose and mannitol with restrained electro static potential (RESP) charge applied to these molecules. The leap.ff14SB.protein force field was employed for GeXIVA[1,2]. We first performed energy minimization for 5000 steps, and then restrained the heavy atoms with a constant force of 1000 kJ/mol/nm^2^ for 100 ps, the final production simulations were performed for 200 ns for the lyophilized product system. The simulation temperature and pressure were set to 300 K with v-rescale coupling method and 1 atmosphere (1.01 bar) with Bredensen coupling method. The hydrogen bond was restrained by the LINC algorithm, which allows us safely to set the time step to 2 fs. All the simulations were performed, and the simulation results were analyzed by GROMACS 2018.

### 4.8. Induction of Chemotherapy-Induced Neuropathic Pain (CINP) and Drugs Administration Procedure

Rats were injected intraperitoneally with freshly prepared paclitaxel (2 mg/kg) after dissolving in normal saline (NS) just before its administration. Paclitaxel was administered on four alternate days (1, 3, 5, and 7 days), as described in a previous study [57]. Rats that developed neuropathic pain symptoms were selected and randomly assigned into three groups (*n* = 6 for each group). Group I served as the control group, which was treated with NS, whereas rats in group II and III were injected with GeXIVA[1,2] (API) and GeXIVA[1,2] lyophilized powder (formulation), respectively. API and formulation were dissolved in normal saline to 0.45 mg/mL, and IM injected at a volume of 0.1 mL/100 g body weight. Mechanical and heat sensitivity was monitored at 0, 1, 2, 4, and 6 h after a single injection. Afterwards, rats received daily repeated administration for 14 days, and the von Frey test and the tail-flick test were performed at 4 h since the last injection. NS was administered according to the same schedule as control. The examiner was blinded to drug treatments.

### 4.9. Behavioral Assessments of Mechanical Allodynia and Heat Hyperalgesia

All manipulations were performed under quiet conditions by the same experimenter in a test room to avoid stress. To test the mechanical sensitivity, animals were placed in suspended cages with a wire mesh bottom and allowed 30 min for habituation before the examination. The plantar surface of each hind paw was stimulated with a series of von Frey hairs of logarithmically incremental stiffness (0.60–26 g, Aesthesio, Danmic, San Jose, CA, USA). These von Frey hairs were presented perpendicularly to the plantar surface (2–3 s for each hair) when the animals were resting, and a positive response was indicated by a sharp withdrawal of the paw. The 50% paw withdrawal threshold (PWT) was determined using Dixon’s up-down method [58]. Heat sensitivity was tested by the tail-flick latency test (TFL) and was measured as the duration of immersion. The tail of the rat was immersed in hot water maintained at 48 °C until withdrawn, with a cut-off time of 15 s to prevent tissue damage [59].

### 4.10. Statistical Analysis

The experimental results were expressed as means ± SEM. The differences between groups were analyzed by one-way variance (ANOVA) followed by a Bonferroni post hoc test. Data were analyzed by SPSS (IBM SPSS Statistics v19.0), and *p* < 0.05 was considered statistically significant. All statistical figures were created using Graphpad Prism 5.0.

## Figures and Tables

**Figure 1 marinedrugs-19-00121-f001:**
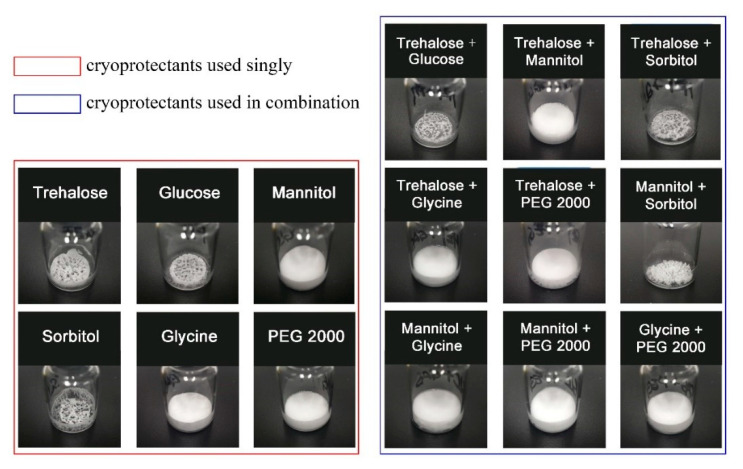
The appearances of the lyophilized products in different formulations. Single cryoprotectant applied in the formulations was shown in the left, while combinations of two cryoprotectants were shown in the right.

**Figure 2 marinedrugs-19-00121-f002:**
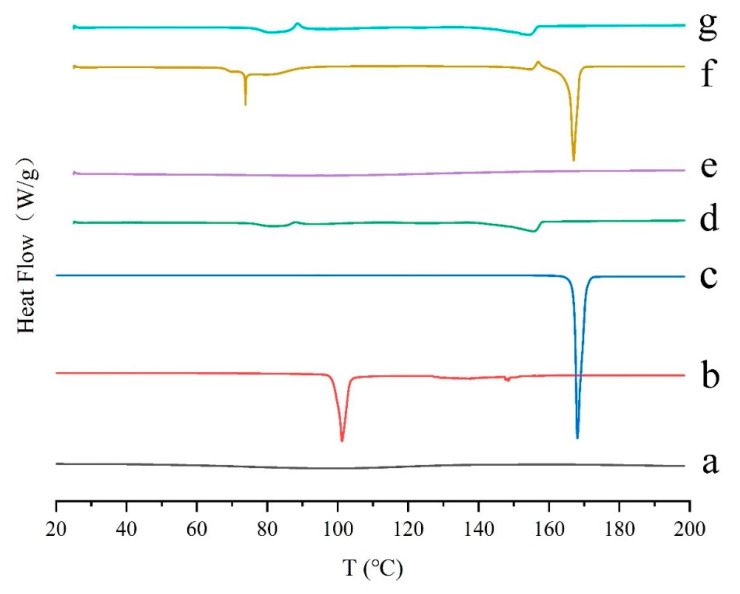
Reversible heat flow of the drug substances and lyophilized powder using differential scanning calorimetry (DSC). Pure GeXIVA[1,2] (**a**), trehalose (**b**), mannitol (**c**), trehalose combined with mannitol (**d**), GeXIVA[1,2] lyophilized with trehalose (**e**), GeXIVA[1,2] lyophilized with mannitol (**f**) and the determined formulation where GeXIVA[1,2] lyophilized with trehalose and mannitol in a combination (**g**) are displayed, respectively.

**Figure 3 marinedrugs-19-00121-f003:**
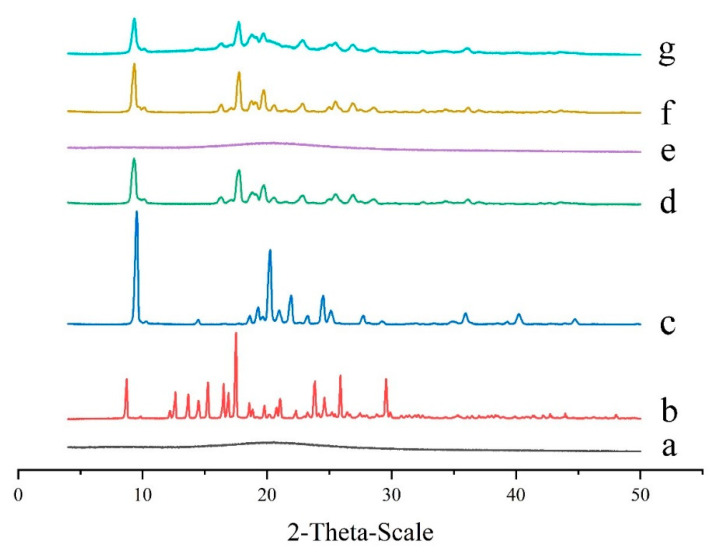
X-ray powder diffraction spectra of pure GeXIVA[1,2] (**a**), trehalose (**b**), mannitol (**c**), trehalose combined with mannitol (**d**), GeXIVA[1,2] lyophilized with trehalose (**e**), GeXIVA[1,2] lyophilized with mannitol (**f**) and the determined formulation where GeXIVA[1,2] lyophilized with trehalose and mannitol in a combination (**g**).

**Figure 4 marinedrugs-19-00121-f004:**
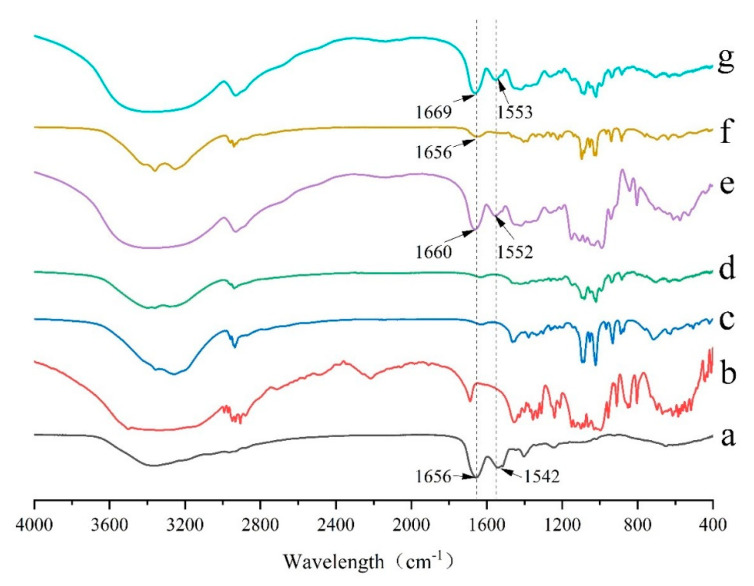
FTIR spectra for different lyophilized samples focused on the amide bonds of pure GeXIVA[1,2] (**a**), trehalose (**b**), mannitol (**c**), trehalose combined with mannitol (**d**), GeXIVA[1,2] lyophilized with trehalose (**e**), GeXIVA[1,2] lyophilized with mannitol (**f**) and the determined formulation where GeXIVA[1,2] lyophilized with trehalose and mannitol in a combination (**g**).

**Figure 5 marinedrugs-19-00121-f005:**
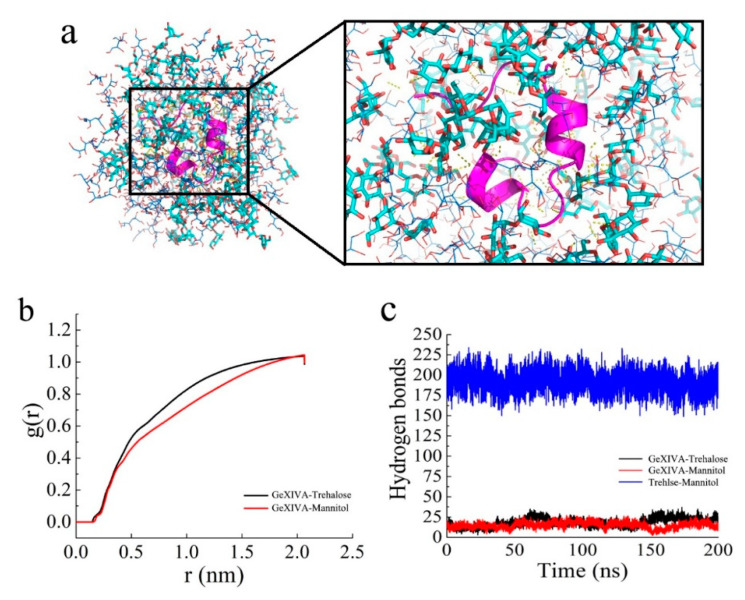
(**a**) The distribution of GeXIVA[1,2] (pink cartoon strip), trehalose (the thicker sticks in light blue), and mannitol (the thinner sticks in deep blue) in the lyophilized products. (**b**) The distribution density of trehalose and mannitol around GeXIVA[1,2]. (**c**) The hydrogen bonds formed in the lyophilized system.

**Figure 6 marinedrugs-19-00121-f006:**
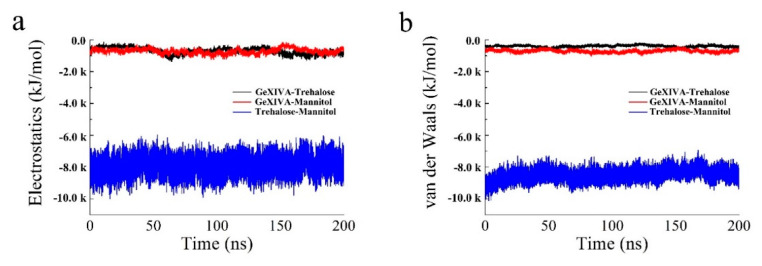
The electrostatic force (**a**) and van der Waals force (**b**) in the lyophilized formulation system.

**Figure 7 marinedrugs-19-00121-f007:**
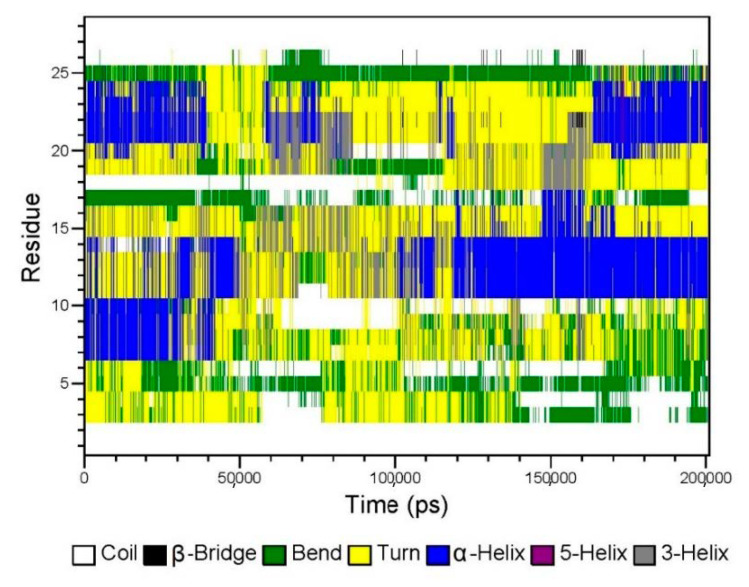
The secondary structure of GeXIVA[1,2] in the lyophilized formulation according to the molecular simulation.

**Figure 8 marinedrugs-19-00121-f008:**
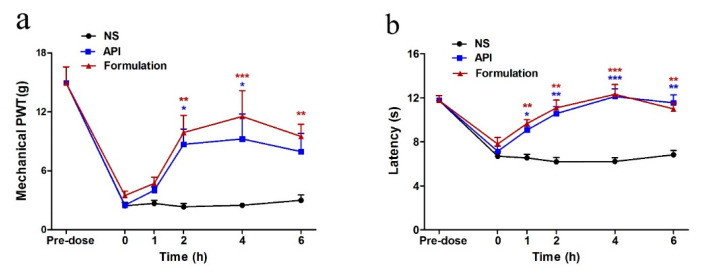
Analgesic effect on paclitaxel-induced neuropathic pain by single intramuscular (IM) injection. (**a**) Time-effect relationship of GeXIVA[1,2] (API) and GeXIVA[1,2] lyophilized powder (formulation) on mechanical paw withdrawal threshold (PWT). (**b**) Time-effect relationship of GeXIVA[1,2] (API) and GeXIVA[1,2] lyophilized powder (formulation) on the tail-flick latency (TFL). Each point indicates the mean ± SEM at the time point, (normal saline (NS), *n* = 6; API, *n* = 6; formulation, *n* = 6). * *p* < 0.05, ** *p* < 0.01, *** *p* < 0.001, compared with the NS control group.

**Figure 9 marinedrugs-19-00121-f009:**
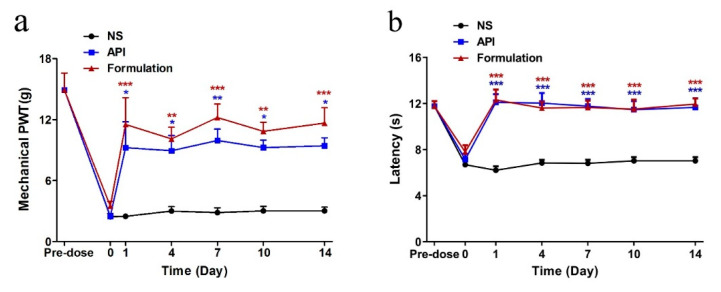
Acute analgesic effect on paclitaxel-induced neuropathic pain following repeated injections. Each data point indicates the mean ± SEM of mechanical PWT (**a**) and TFL (**b**) at 4 h post-injection (*n* = 6,). * *p* < 0.05, ** *p* < 0.01, *** *p* < 0.001, compared with the NS group.

**Table 1 marinedrugs-19-00121-t001:** Forced degradations of GeXIVA[1,2] in various stress conditions.

Stress Conditions	Concentration of Stressor	Duration	Assay (% *w*/*w*)
None	H_2_O	5 days	98.74 ± 0.12
Acid	0.1 M HCl	48 h	97.59 ± 0.23
Alkali	0.01 M NaOH	2 min	88.22 ± 0.95
Oxidation	3% H_2_O_2_	12 h	89.35 ± 0.48
Photolysis	4500 ± 500 Lx	5 days	97.46 ± 0.21
		10 days	95.37 ± 0.32
Thermal	40 °C dry heat	5 days	80.53 ± 0.98
	60 °C dry heat	5 days	23.79 ± 1.42
pH	pH 4.0	24 h	98.02 ± 0.26
	pH 5.0		98.28 ± 0.31
	pH 6.8		99.27 ± 0.13
	pH 7.4		97.56 ± 0.28
	pH 8.0		85.36 ± 0.42
	pH 9.0		60.42 ± 0.86

**Table 2 marinedrugs-19-00121-t002:** Assay of GeXIVA[1,2] in different formulations after lyophilization process and stored under accelerating condition.

Lot	Cryoprotectors	Assay (% *w*/*w*)			
		0 Day	Accelerating Condition—5 Days	Accelerating Condition—10 Days	Accelerating Condition—30 Days
1	Trehalose	97.03 ± 0.49	98.33 ± 0.02	97.31 ± 0.62	97.14 ± 0.77
2	Glucose	50.87 ± 1.96	16.70 ± 1.51	10.89 ± 0.20	10.88 ± 0.20
3	Mannitol	96.48 ± 1.81	10.10 ± 0.19	10.63 ± 5.66	N.A
4	Sorbitol	96.73 ± 0.67	90.12 ± 0.62	70.95 ± 4.66	54.70 ± 2.57
5	Glycine	94.20 ± 1.37	80.14 ± 0.49	78.33 ± 1.64	67.20 ± 1.29
6	PEG 2000	88.42 ± 1.33	82.13 ± 0.51	71.43 ± 0.22	56.99 ± 0.04
7	Trehalose + Glucose	39.56 ± 7.89	14.46 ± 3.78	11.87 ± 0.50	N.A
8	Trehalose + Mannitol	98.49 ± 0.45	98.02 ± 0.08	98.42 ± 0.26	98.36 ± 0.24
9	Trehalose + Sorbitol	99.02 ± 0.58	94.72 ± 0.04	90.76 ± 0.37	79.65 ± 0.04
10	Trehalose + Glycine	98.38 ± 0.38	95.82 ± 0.18	96.49 ± 1.29	85.13 ± 1.39
11	Trehalose + PEG 2000	97.69 ± 1.45	99.28 ± 0.36	91.53 ± 4.25	68.76 ± 4.29
12	Mannitol + Sorbitol	97.17 ± 1.31	67.37 ± 7.18	40.51 ± 0.11	26.49 ± 0.19
13	Mannitol + Glycine	97.30 ± 0.23	96.40 ± 1.13	93.43 ± 0.98	80.50 ± 0.17
14	Mannitol + PEG 2000	93.47 ± 0.14	78.50 ± 0.03	66.74 ± 0.89	43.98 ± 0.78
15	Glycine + PEG 2000	99.14 ± 4.38	84.49 ± 0.62	74.08 ± 1.32	51.37 ± 1.23

## Data Availability

The data presented in this study are available on request from the corresponding author. The data are not publicly available due to some privacy issues about drug development.

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
