# Peer review of "Lyophilization Serves as an Effective Strategy for Drug Development of the α9α10 Nicotinic Acetylcholine Receptor Antagonist α-Conotoxin GeXIVA[1,2]"

_marinedrugs, 2021, doi:10.3390/md19030121_

Round 1
Reviewer 1 Report
Review of Marine Drugs 1121153
This is a useful manuscript and is quite well written. My recommendations are below.
Abstract
This should be revised in line with any revisions suggested and made in the main text
Introduction
Line 42 Citations [11-15] do not include references to schizophrenia or ADHD, as in the text. A relevant citation should be included.
Lines 49-50 Citation needed
Line 56 …”which is expected to bring the gospel”… this should be rephrased
Lines 59-60 Citation needed
Line 70 Citation 32 is incomplete
Lines 74-75 Citation needed
Materials and Methods
Line 402 Citation needed
Section 4.4 Multiple citations needed
Line 431 Should this be 2.5% excipients, not 5% ?
Line 432 More details are needed about the vials and stoppers used (supplier, type) especially since the vials can influence product stability
Line 435 Should this be 20 millibar ?
Lines 437-438 Were the vials fitted with overseals after freeze drying ?
Line 457 20psi should be expressed in metric units
Line 476 GAFF and RESP should be written in full
Line 477 Filed?
Line 481 1atm should be expressed in metric units
Line 491 “normal” should be “control”. The authors should explain why they did not use a true control – the same formulation as the product but without the conotoxin component
Results
Table 1 The authors should report why they chose such extreme forced degradation conditions and what is the relevance to normal storage and preservation of the conotoxin. For example, why 3% H2O2, why not 0.3%? The product will never see such extreme conditions. How does the photodegradation study compare with ICH recommendations? What were the pH values for the acid and alkali conditions?
Line 108 Appearance has nothing to do with patient acceptance. The vast majority of patients being treated with the product will never see the lyophilised vials.
Lines 109-112 I disagree that these 2 factors are the main cause for change in the assay. The presence of incompatible excipients may be a major factor as well. Also issues about incompatibility of the excipients with the assay methodology must be taken into account.
Line 117 There is no need to have both Table 2 and Figure 2 with the same data. I recommend removing Figure 2, the table is much easier to follow.
Lines 117-118, 129 The phrase “had good protective effects on the assay” is strange and actually not correct. Those excipients had good protective effects on the conotoxin molecule, not the assay.
Line 121 “great” should be “good”
Lines 123-125 Need rewording
Lines 133-134 Need rewording
Lines 150-152 Presumable this was with trehalose and mannitol?
Lines 152-154 Citation from regulatory authorities needed
Lines 160-162 Citations needed
Line 164 “which nicely meet”… should be reworded
Lines 174-176 Citation needed together with some brief explanation of how DSC results should be interpreted
Line 197 To check that these are the correct citations here ?
Figure 4 The authors should comment about 4(f) and 4 (g) being almost identical
Line 229 “worse still” should be reworded
Lines 232-233 The authors should comment on the amide bond peak shift in the final formulation
Figure 6A It is very difficult to see the differences between light blue and deep blue in such a small figure……
Line 252 Figure 6(b) ?
Lines 258 259 Figure 6(c) ?
Line 287 Figures 9 (a) and (b) both already show an effect at 1 hour post-treatment, not 2 hours
Section 2.5.3 It is a shame that the authors did not investigate just how long a single dose treatment of the conotoxin could last, rather than a repeat dose study
Discussion
Line 310 To note, Prialt is a synthetic peptide not the naturally-occurring conotoxin
Line 340 “great success” should be reworded
Reviewer 2 Report
The manuscript entitled “lyophilization serves as an effective strategy for drug development of the a9a10 nicotinic acetylcholine receptor antagonist a-Conotoxin GeXIVA[1,2]” submitted by Dr Zhiguo Li and colleagues is well written and easy to read. The experimental design is robust and detailed. Findings are thoroughly described. In my opinion, the manuscript can be accepted for publication in the present status.
Author Response
We are honored to hear from you about this, and thank you very much for your affirmation of our work. We will go on without a break to devote into the research work.
Reviewer 3 Report
Comments to Author:
1) In the manuscript authors has done the Drug Development of the α9α10 Nicotinic Acetylcholine Receptor Antagonist α-Conotoxin GeXIVA[1,2]. Did author look for another nAChRs.
2) The author has not explained how much cost effect this method as compare to the normal drug administrative process because as authors mention sometime these peptides can get degraded.
I would suggest the author include those to make the study more interesting
Author Response
- point 1
To be honest, we just do some research focused on the α9α10 Nicotinic Acetylcholine Receptor. We have not searched for another nAChRs till now.
- point 2
Through the freeze-drying technology, the stability of the product is greatly improved, and the drug products can be stored at room temperature for about 9-12 months, which is almost impossible for the normal drug administrative process since the peptides would get degraded in solution for a few days.

Round 2
Reviewer 1 Report
The authors have made good revisions to their manuscript, which is improved significantly.